# Rare *VPS35* A320V Variant in Taiwanese Parkinson’s Disease Indicates Disrupted CI-MPR Sorting and Impaired Mitochondrial Morphology

**DOI:** 10.3390/brainsci10110783

**Published:** 2020-10-27

**Authors:** Yih-Ru Wu, Chih-Hsin Lin, Chih-Ying Chao, Chia-Wen Chang, Chiung-Mei Chen, Guey-Jen Lee-Chen

**Affiliations:** 1Department of Neurology, Chang Gung Memorial Hospital, Linkou Medical Center and Chang Gung University College of Medicine, No. 5, Fusing St., Gueishan Township, Taoyuan County 33302, Taiwan; yihruwu@cgmh.org.tw (Y.-R.W.); chli416@hotmail.com (C.-H.L.); b86206029@ntu.edu.tw (C.-Y.C.); chiawen0108@gmail.com (C.-W.C.); 2Department of Life Science, National Taiwan Normal University, 88 Ting-Chou Road, Section 4, Taipei 11677, Taiwan

**Keywords:** Parkinson’s disease, *VPS35*, mutation screen, CI-MPR sorting, impaired mitochondrial morphology

## Abstract

Sequence variants in vacuolar protein sorting 35 (*VPS35*) have been reported to be associated with Parkinson’s disease (PD). To investigate if the genetic variants in *VPS35* contribute to Taiwanese PD, *VPS35* cDNA fragments from 62 patients with PD were sequenced. A cohort of PD (*n* = 560) and ethnically matched controls (*n* = 506) were further examined for the identified mutation. The effects of the mutation on cation-independent mannose-6-phosphate receptor (CI-MPR) sorting and mitochondrial morphology were further examined in 293T cells expressing the mutant *VPS35*. Here, a novel heterozygous A320V in the *VPS35* gene was identified in two late-onset PD (LOPD) patients, which was absent in 506 normal controls. Expression of the A320V mutant in 293T cells demonstrated increased colocalization of *VPS35* with CI-MPR and decreased CI-MPR and lysosomal-associated membrane protein 2 (LAMP2) levels. Decreased CI-MPR manifested in missorting of cathepsin D and decreased proteolysis of α-synuclein. A320V mutation also increased mitochondrial E3 ubiquitin protein ligase 1 (MUL1) and thus led to mitofusin 2 (MFN2) degradation. The results suggest that the expression of *VPS35* A320V leads to disrupted CI-MPR sorting and impaired mitochondrial morphology, which may partly explain its action in PD.

## 1. Introduction

Parkinson’s disease (PD) is the second most common neurodegenerative disease, and its pathogenesis is still unknown. A complex interaction between genetic and environmental factors contributes to development of this disease [1]. Genetic research in PD has identified mutations in several genes, including *SNCA*, *LRRK2*, and *VPS35* or *PARKIN*, *PINK1*, *DJ-1*, *ATP13A2*, *PLA2G6*, and *FBXO7*, which are inherited in an autosomal dominant or autosomal recessive manner, respectively [2]. Furthermore, common genetic variations at the *SNCA*, *LRRK2*, and *GBA* loci confer risk for developing idiopathic PD [2]. Newly reported genes are *DNAJC13*, *CHCHD2*, and *TMEM230* for dominant PD and *SYNJ1*, *RAB39B DNAJC6*, *VPS13C*, and *PTRHD1* for recessive PD [3]. However, the pathogenicity of some newly reported genes (*DNAJC13*, *CHCHD2*, and *TMEM230*) remains to be validated. The abovementioned gene products have been reported to be involved in regulating several cellular pathways, including mitochondrial turnover, synaptic vesicle exocytosis/endocytosis, endosomal sorting, autophagy and lysosomal function, and neuroinflammation [4].

Mutations of vacuolar protein sorting 35 (*VPS35*, PARK17, OMIM: 614203) gene in PD have been explored (review in [5]). Initially, Vilarino-Guell and coworkers identified an aspartic-acid-to-asparagine mutation at residue 620 (p.D620N, c.1858G>A) in a Swiss family, three other families, and one patient with sporadic PD, while a proline-to-serine variant was identified at residue 316 (p.P316S, c.946C>T) in the *VPS35* gene in a United States family in 2011 [6]. Following this initial report, other groups also reported the D620N mutation in a number of individuals and families with PD worldwide [5,7]. With the exception of Japanese [8], D620N mutation is predominantly found in families of Caucasian descent with autosomal dominant PD. The frequency of the D620N mutation in patients with familial PD is estimated to be 0.1% to 1% [9]. In addition, several rare variants have been identified in familial PD or sporadic PD cases, such as R32S, P51S, R524W, I560T, H599R, M607V, and L774M (review in [5]). However, the pathogenicity of most of these variants remains inconclusive.

The *VPS35* protein functions as a core of the retromer complex that facilitates both endosome-to-plasma membrane and endosome-to-Golgi complex transport as well as recycling of transmembrane protein cargo [10]. *VPS35* in the cargo recognition *VPS26–VPS29–VPS35* complex binds the cation-independent mannose 6-phosphate receptor (CI-MPR) in endosomes and transports CI-MPR to the *trans*-Golgi network (TGN) to prevent its lysosomal degradation [11]. CI-MPR at the TGN binds mannose 6-phosphate groups on the lysosomal hydrolase precursors to transport lysosomal enzymes to lysosomes [12]. In a transgenic *Drosophila* model expressing human α-synuclein (SNCA), depletion of *VPS35* resulted in inappropriate trafficking of CI-MPR and cathepsin D (CTSD), leading to enhanced α-synuclein accumulation and exacerbated locomotor impairments in flies [13]. Therefore, *VPS35* deficiency has been suggested to increase α-synuclein accumulation via impaired lysosomal function.

The pathogenesis of the reported *VPS35* P316S, R524W, and D620N mutations has been investigated. Utilizing *Drosophila* as a model, D620N and, to a smaller extent, P316S led to late-onset loss of dopaminergic (DA) neurons, poor mobility, reduced survival, and increased sensitivity to PD-linked rotenone toxin [14]. The endosomal trafficking of CI-MPR and subsequent CTSD processing was disrupted by the D620N mutation in model cell lines and PD fibroblasts with D620N mutation [15]. Additionally, DA neurons expressing PD-linked D620N impaired endosome-to-Golgi retrieval of lysosome-associated membrane protein 2 (LAMP2), which causes increased α-synuclein accumulation [16]. Moreover, PD-linked P316S, R524W, and D620N mutants increased mitochondrial E3 ubiquitin protein ligase 1 (MUL1) levels and thus led to mitofusin 2 (MFN2) degradation and mitochondrial fragmentation [17]. Mitochondrial fission and neuronal loss were also observed by enhanced dynamin-like protein 1 (DLP1) complex turnover, with more pronounced effects induced by D620N compared to R524W [18].

To date, reported evidence on the pathogenicity of rare *VPS35* mutations is limited. This study aims to investigate if the genetic variants in *VPS35* are etiologically relevant to the pathogenesis of Taiwanese PD.

## 2. Materials and Methods

### 2.1. Ethics Statement

This study was performed under a protocol approved by the institutional review boards of Chang Gung Memorial Hospital (No. 101-5299A3), and all examinations were performed after obtaining written informed consents.

### 2.2. Subjects

Patients diagnosed with PD (onset age 19–93 years, average age 62.9 ± 11.2 years, 45.7% females, *n* = 622) were recruited from the neurology clinics of Chang-Gung Memorial Hospital. Among them, a subgroup of PD (onset age 51–78 years, average age 59.6 ± 7.5 years, 45.2% females, *n* = 62) were randomly selected for *VPS35* cDNA sequencing. The diagnosis of PD was based on the UK PD Society Brain Bank clinical diagnostic criteria [19]. Unrelated healthy adult volunteers (59.8 ± 12.8 years, 52.4% females, *n* = 506) matched for age, gender, ethnic origin, and area of residence were recruited as controls. To avoid the skew caused by multiple family members carrying the same genetic variants, we only included one proband if patients had family history of PD.

### 2.3. Genetic Analysis

Genomic DNA and RNA were extracted from leukocytes using the standard protocols. RNA from the abovementioned 62 PD patients was reverse transcribed to cDNA (SuperScript^TM^ III reverse transcriptase; Invitrogen, Carlsbad, CA, USA) and PCR amplified (forward primer: 5′-ATGACCGCGGGAGGCTAC, reverse primer: 5′-CCGGAGGTGCTGGGTAAAAC) for *VPS35* cDNA direct sequencing. The identified *VPS35* A320V variant was screened in 506 controls and the rest of the 560 PD patients by genomic DNA PCR (forward primer: 5′-GAGTGGAGAGTGCGTGACTG, reverse primer: 5′-AACAAGACAAATGCCAGCAG) and *Bse*YI (loss of site) restriction analysis. We also screened the reported D620N variant in all the recruited PD patients by genomic DNA PCR (forward primer: 5′-CTCAGTTGTGTCCTGTGGCTCTC, reverse primer: 5′-CCTCCTCCCCATTTTTGTCC) and *Hin*fI (gain of site) restriction analysis.

### 2.4. VPS35 and SNCA cDNA Constructs

The *VPS35* cDNA (NM_018206) in pOTB7 was obtained from Bioresource Collection and Research Center (BCRC), Food Industry Research and Development Institute, Taiwan. The *VPS35*-containing *Bam*HI/*Bsa*I fragment, a linker flanked by *Bsa*I and *Nhe*I sites (generated by annealing primers 5′-TCCTTGAAAAAGGATCCCCG and 5′-CTAGCGGGGATCCTTTTTCA), and ZsYellow1-containing *Nhe*I/*Dra*I fragment (from pZsYellow1-C1) were subcloned into *Bam*HI- and *Eco*RV-digested pcDNA5/FRT/TO (Novagen, Madison, WI, USA) to generate pcDNA5-*VPS35*-ZsYellow1. The resulting construct had a 12 amino acid linker (EKGSPLALPVAT) between *VPS35* and ZsYellow1. To clone A320V *VPS35*, cDNA amplified from a late-onset PD (LOPD) patient carrying heterozygous A320V (patient H1914) was cloned into pGEM-T Easy (Promega, Madison, WI, USA) and sequenced. Then, DNA fragment containing A320V was excised with *Eco*NI-*Afl*II and subcloned into the corresponding sites in pcDNA5-*VPS35*-ZsYellow1. The fidelity of the cDNA constructs was confirmed by restriction digestion and/or direct nucleotide sequencing.

Polyadenylated RNA (200 ng) isolated from neuroblastoma SK-N-SH cells was reverse transcribed as described. The sense and antisense primers for α-synuclein (*SNCA*, NM_000345) cDNA amplification were 5′-GCGGCCGCCATGGATGTATTCATGAAAGG (forward, *Not*I site underlined) and 5′-AGATCTGGCTTCAGGTTCGTAGTCTTG (reverse, *Bgl*II site underlined). The amplified *SNCA* cDNA was cloned into pGEM-T Easy vector (Promega) and sequenced. Then, the *Not*I/*Bgl*II fragment containing *SNCA* cDNA and *Bam*HI/*Xho*I fragment containing Myc epitope and polyhistidine tag (from pcDNA3.1/myc-his, Invitrogen) were subcloned into *Not*I- and *Xho*I-digested pcDNA3 vector (Invitrogen) to generate pcDNA3-SNCA-Myc-His.

### 2.5. Cell Cultivation and Transfection

Human embryonic kidney (HEK)-293T (ATCC No. CRL-11268) cells were grown in a humidified 37 °C incubator with 5% CO_2_ atmosphere and maintained in Dulbecco’s modified Eagle’s medium containing 10% fetal bovine serum (FBS). Cells were plated into six-well (6 × 10^5^/well) dishes, grown for 20 h, and cotransfected with *VPS35*-ZsYellow1 (wild-type and A320V) and SNCA-His constructs (2.5 µg each) into cells using Lipofectamine 2000 according to the manufacturer’s instructions (Life Technologies, Carlsbad, CA, USA). The cells were grown for 48 h for the protein studies.

### 2.6. Western Blotting

Cells were lysed in lysis buffer (20 mM HEPES pH 7.4, 1 mM MgCl_2_, 10 mM KCl, 1 mM DTT, 1 mM EDTA pH 8.0) containing the protease inhibitor mixture (Calbiochem, San Diego, CA, USA). After sonication and sitting on ice for 20 min, the lysates were centrifuged at 14,000× *g* for 30 min at 4 °C. Total proteins (25 µg) were electrophoresed on 10% sodium dodecyl sulfate (SDS)-polyacrylamide gel and blotted onto nitrocellulose membranes. After blocking, the membrane was probed with antibodies against CI-MPR (1:20000; Abcam, Cambridge, MA, USA), *VPS35* (1:1000; GeneTex, Irvive, CA, USA), CTSD (1:500; Santa Cruz Biotechnology, Santa Cruz, CA, USA), LAMP2 (1:1000; Abcam, Cambridge, MA, USA ), MUL1 (1:1000; Abcam, Cambridge, MA, USA), MFN2 (1:1000; Cell Signaling, Danvers, MA, USA), His (1:1000; Acris Antibodies GmbH, Herford, Germany), β-actin (ACTB, 1:5000; Millipore, Billerica, MA, USA), and glyceraldehyde-3-phosphate dehydrogenase (GAPDH, 1:1000; MDBio, Taipei, Taiwan) at 4 °C overnight. Horseradish peroxidase-conjugated goat antimouse or goat antirabbit IgG antibody (1:5000; GeneTex, Irvive, CA, USA) and chemiluminescent substrate (Millipore) were used to detect the immune complexes.

### 2.7. Immunocytochemistry and Confocal Microscopy Examination

HEK-293T cells were plated into 12-well dishes (on coverslips, 2 × 10^5^/well), grown for 20 h, and transfected with the wild-type and A320V *VPS35*-ZsYellow1 plasmids (2 µg each) for 48 h. Cells were then washed with phosphate-buffered saline (PBS) and fixed in 4% paraformaldehyde in PBS for 15 min, followed by 20 min incubation with 0.1% Triton X-100 in PBS to permeabilize cells and an overnight incubation with 2% bovine serum albumin (BSA) in PBS to block nonspecific binding. The primary CI-MPR (1:2000; Abcam) antibody was used to stain cells at 4 °C overnight. After washing with PBS containing 0.1% Tween 20 (PBST), cells were incubated for 2 h at room temperature in Cy5-conjugated secondary antibody (1:500; Invitrogen, Carlsbad, CA, USA) and washed in PBST. Nuclei were detected using 4′-6-diamidino-2-phenylindole (DAPI). Cells were examined after mounting on Vectashield (Vector Laboratories, Inc., Burlingame, CA, USA) for ZsYellow1 and Cy5 fluorescence using a Zeiss LSM 880 confocal laser scanning microscope (Carl Zeiss Microscopy, Oberkochen, Germany) optimized for simultaneous fluorescent imaging. For colocalization, 60 cells from three independent experiments were analyzed and quantified according to the proportion of pixels that were both red and green using Pearson’s rank coefficient as part of the ImageJ Coloc_2 plugin (rsb.info.nih.gov/ij/).

For mitochondrial morphology analysis, wild-type and A320V *VPS35*-ZsYellow1-transfected cells were stained for 20 min at 37 °C with 200 nM MitoTracker Deep Red FM (Invitrogen). The cells were then fixed, permeabilized, washed, and nuclei stained, and images were acquired as described above. Image analysis was performed using ImageJ macro tool to define the perimeter of mitochondria and calculate aspect ratio (long axis/short axis) and form factor (1/circularity, perimeter^2^/4 π × area). Mitochondria morphology analysis was repeated three times with at least 100 cells in each time of measurement.

### 2.8. Statistical Analysis

For statistical analysis of immunoblots and microscopy images, data were expressed as the means ± standard deviation (SD). Experiments in triplicate were performed, and noncategorical variables were compared using the Student’s *t*-test. All *p* values were two-tailed, with values of *p* < 0.05 being considered significant.

## 3. Results

### 3.1. Mutation Analysis

*VPS35* cDNA fragments from 62 PD patients were amplified for sequence analysis. A novel heterozygous substitution leading to an amino acid change from alanine to valine at position 320 (p.A320V) was identified in a LOPD patient (H1914) (Figure 1A). A320V can be differentiated using PCR and *Bse*YI restriction analysis (Figure 1B). A320V, a nonpolar amino acid residue located between *VPS35* α-helices 14 and 15 [20], is evolutionary conserved in the known mammalian homologues of the *VPS35* protein (Figure 1C). A320V was found in another LOPD patient (H1967) upon PCR and *Bse*YI restriction analysis to screen the rest of the 560 patients, but it was not found in the 506 controls. In addition, known D620N was not found in any of our PD patients.

### 3.2. Clinical Analysis of Patients with A320V

There were two patients carrying A320V in our cohort. The first one (H1914) was a 67-year-old female, presented with right hemiparkinsonian features without other parkinsonism-plus abnormalities, which is identical to idiopathic PD at age 55. She had a very good response to levodopa with independent daily activity for five years. She started to have end-dose phenomenon to levodopa treatment seven years after onset of PD. Unfortunately, she got right parietal and occipital subcortical infarction at the age of 65, which made her confined to a wheelchair. Computer tomography of the brain performed at age 60 was normal. There were no other family members suffering from the same symptoms. The second one (H1967) came to our clinic at age 68. She had rest tremor and was clumsy of her right upper limb at age 67. PD was diagnosed according to the excellent response to levodopa as well as normal magnetic resonance imaging of the brain and an asymmetric reduction of 99mTc-TRODAT-1 uptake in the left striatum compared with a normal individual. She had no other family members or relatives having the similar symptoms.

### 3.3. Impaired Endosome-to-Golgi Retrieval of CI-MPR in A320V-Expressing Cells

CTSD is the main lysosomal hydrolase responsible for the degradation of α-synuclein [21]. In human cells, CTSD is first synthesized as pre-pro-CTSD (412 amino acids), which is processed to pro-CTSD (394 amino acids) by the removal of the signal peptide in the endoplasmic reticulum. After transport to the late endosomes and lysosomes, pro-CTSD is converted to mature disulfide-linked heavy (244 amino acids) and light (97 amino acids) chains. To examine if A320V affected *VPS35* retromer function, the *VPS35*-ZsYellow1 and SNCA-Myc-His cDNAs were co-overexpressed in HEK-293T cells (Figure 2A and Appendix A). When compared to wild-type *VPS35*-transfected cells, the amount of CI-MPR and matured CTSD heavy chain proteins in A320V mutant-transfected cells were significantly reduced (CI-MPR: 81% vs. 100%, *p* = 0.015; CTSD: 83% vs. 100%, *p* = 0.020), suggesting impaired delivery of pro-cathepsin D to the late-endosome/lysosome for processing into the mature 27 kDa form. Notably, these changes were accompanied by a concomitant increase of SNCA detected by a His-tag antibody (131% vs. 100%, *p =* 0.002).

Additionally, the expression level of LAMP2 isoform A (LAMP2A) was examined. As shown in Figure 2A and Appendix A, the amount of LAMP2A in A320V mutant-transfected cells was significantly reduced (65% vs. 100%, *p* = 0.004), suggesting accelerated degradation of LAMP2A.

To determine if the presence of *VPS35* A320V mutation impacts retromer-mediated trafficking of CI-MPR, studies in colocalization between transfected *VPS35* and CI-MPR in HEK-293T cells were performed. Colocalization was significantly increased for CI-MPR and A320V (26% vs. 17%, *p* = 0.038) (Figure 2B) compared with that found in wild-type-transfected cells. Together with the observed reduced amounts of CI-MPR, our study results may suggest impaired endosome-to-Golgi retrieval of CI-MPR in *VPS35* A320V-expressing cells.

### 3.4. Increased MUL1 and Reduced MFN2 in A320V-Expressing Cells

It is thought that mitochondrial fusion morphology is critical for maintenance of mitochondrial respiratory capacity in mammalian cells [22]. Given that *VPS35* is a critical regulator for mitochondrial E3 ubiquitin ligase MUL1 and its substrate MFN2, we examined whether *VPS35* A320V mutant affected MUL1 level, resulting in MFN2 degradation and mitochondrial fragmentation. As shown in Figure 3A and Appendix A, the amount of MUL1 protein in A320V mutant-expressing cells was significantly increased (134% vs. 100%, *p* = 0.004), which was associated with a decrease of MFN2 (74% vs. 100%, *p* = 0.001). Accordingly, the average aspect ratio (1.9 vs. 2.2, *p* = 0.033) and form factor (5.8 vs. 7.3, *p* = 0.008) were significantly lower in A320V-expressing cells when compared to wild-type *VPS35*-transfected cells, demonstrating that A320V may induce small and round mitochondria, indicative of mitochondrial fragmentation (Figure 3B). 

## 4. Discussion

In the present study, we investigated *VPS35* mutation in Taiwanese PD patients. One heterozygous variant, A320V, was identified in two LOPD patients among the 610 patients screened (allele frequency: 0.16%). A320V has been reported in the NCBI single nucleotide polymorphisms (dbSNP) database (rs747944333), with allele frequency of 0.015% in a study of 516,606 chromosomes. Based on Exome Aggregation Consortium (ExAC), this variant was mostly found in Latino (0.1%), while it was very rare in Asian (0.004%). Furthermore, this variant has not been reported in the Taiwan Biobank (https://taiwanview.twbiobank.org.tw/). Although p.D620N mutation has been reported in Taiwan [7]; however, this mutation was not found in our PD patients. Our results suggested that *VPS35* mutation may not be a common cause of Taiwanese PD. This is in accordance with those previously reported [23,24,25].

*VPS35* is the largest of the cargo recognition complex. Analysis of the *VPS35* sequence showed that it is composed of a total 34 helices (17 two-helix repeats) that fold into an α-solenoid, with 13 C-terminal helices wrapping around VPS29 and 8 N-terminal helices interacting with VPS26 [20]. A320V is located between *VPS35* α-helices 14 and 15, thus more likely to not affect the formation of VPS26–VPS29–*VPS35* heterotrimer. Most retromer cargoes possess at least one simple hydrophobic motif, F/W-L-M/V, required for retromer-dependent sorting [26], and *VPS35* has long been considered to provide the sole interface for cargo recognition. To understand the possible underlying molecular mechanism of *VPS35* A320V mutation, we examined if trafficking of CI-MPR was disrupted by expression of this variant. Within HEK-293T cells expressing A320V, increased colocalization of A320V with CI-MPR (Figure 2B) may implicate reduced retrieval of CI-MPR to TGN, which may lead to accelerated lysosomal degradation of CI-MPR (Figure 2A). An increased dispersal of CI-MPR was also observed in different patient cells with D620N mutation [27]. Whether reduced retrieval of LAMP2A leads to decreased LAMP2A (Figure 2A) remains to be determined. LAMP2 is a receptor for chaperon-mediated autophagy (CMA) [28]. Decreased LAMP2A may impair CMA-mediated α-synuclein degradation, given the critical role of LAMP2A in CMA [29]. In addition, the *VPS35* A320V variant reduces the retromer-mediated trafficking of cathepsin D, leading to decreased maturation of cathepsin D, which causes reduced degradation of α-synuclein in the lysosome (Figure 2A). However, whether *VPS35* A320V promotes degradation of CI-MPR and LAMP2A by lysosome warrants substantial evidence from further studies. The observed endosomal and lysosomal alterations and trafficking defects resulting from A320V may have a strong correlation to the development and progression of PD.

Similar to A320V, the reported P316S is also located between *VPS35* α-helices 14 and 15. Transgenic fly studies have demonstrated that expression of P316S leads to the loss of tyrosine hydroxylase-positive dopamine neurons, locomotor dysfunction, and reduced lifespan, although to a smaller extent [13]. Expression of P316S is also sufficient to increase MUL1 and reduce MFN2 [16]. Similar to P316S, *VPS35* A320V mutant may affect mitochondrial dynamics by increasing mitochondrial MUL1, which leads to MFN2 degradation (Figure 3A) and results in mitochondrial fragmentation. While we showed that A320V altered expression of MUL1 and MFN2, further studies are required to demonstrate if A320V causes mitochondrial abnormalities, including fragmentation.

Familial PD-associated *VPS35* mutants can cause mitochondrial fragmentation through enhanced *VPS35*-DLP1 interaction to recycle DLP1 complexes, eventually leading to mitochondria dysfunction and neuronal loss [18], and D620N mutant-induced mitochondrial fragmentation and respiratory deficits could be rescued by blocking the *VPS35*-DLP1 interaction and inhibiting the recycling of mitochondrial DLP1 complexes [30]. A320V may affect DLP1 complex recycling, resulting in the observed small and round mitochondria in A320V-expressing cells (Figure 3B), but this remains to be clarified by future studies.

Although increased α-synuclein accumulation via impaired lysosomal function in *VPS35* knock-down *Drosophila* model has been shown [13], a recent study displayed robust tau pathology in D620N *VPS35* knock-in (KI) mouse model, with no evidence of α-synuclein-positive neuropathology in aged *VPS35* KI mice [31]. In addition to the difference in model organism used, the discrepancy of findings may be in part related to the different pathogenicity of *VPS35* knock-down and D620N point mutation as D620N has been reported to result in a gain of function, causing PD through hyperactivation of the LRRK2 kinase [32] and alteration of the dopaminergic system [33].

Although our findings emphasize a possible vulnerability of developing PD in those carrying the *VPS35* A320V variant, our observation may be biased as functional studies were performed after overexpression of the wild-type or A320V *VPS35* and SNCA, and the observed effects could be rather divergent from effects caused by endogenous physiological expression levels of *VPS35* or SNCA. The functional characteristics of *VPS35* A320V need to be further validated using fibroblasts or neurons derived from *VPS35* A320V carriers. Another limitation of our study is that mutations in recently reported genes, such as *SYNJ1*, *RAB39B*, *DNAJC6*, *VPS13C*, and *PTRHD1*, were not excluded in our PD patients sequenced for *VPS35* cDNA. As cDNA sequencing may miss intronic/splice site variants, further analyses of the *VPS35* gene on genomic DNA and functional studies are also warranted to determine *VPS35* genetic variants contributing to the risk of Taiwanese PD.

## 5. Conclusions

We identified *VPS35* A320V mutation in two LOPD from a cohort of 622 PD patients. A320V increased colocalization with CI-MPR and was associated with decreased CI-MPR and LAMP2 expression. Decreased CI-MPR manifested in missorting of cathepsin D, leading to decreased α-synuclein degradation. In addition, A320V increased MUL1, enhanced MFN2 degradation, and impaired mitochondrial morphology. Although *VPS35* mutations are a rare cause of PD in Taiwan, the identification and characterization of the *VPS35* A320V variant contributes to the understanding of *VPS35*-induced PD.

## Figures and Tables

**Figure 1 brainsci-10-00783-f001:**
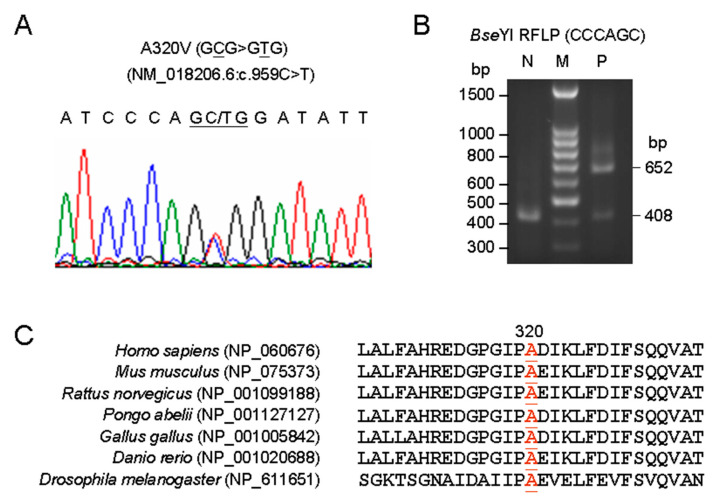
Mutation identification. (**A**) Chromatograms of direct cDNA sequencing of A320V. (**B**) PCR and *Bse*YI restriction analysis of A320V. PCR-amplified products from a patient (lane P) and a normal control (lane N) were digested with *Bse*YI and resolved on a 1.4% agarose gel. Lane M contains size markers. (**C**) Evolutionary conservation of the regions of vacuolar protein sorting 35 (*VPS35*) A320V using the program Vector NTI.

**Figure 2 brainsci-10-00783-f002:**
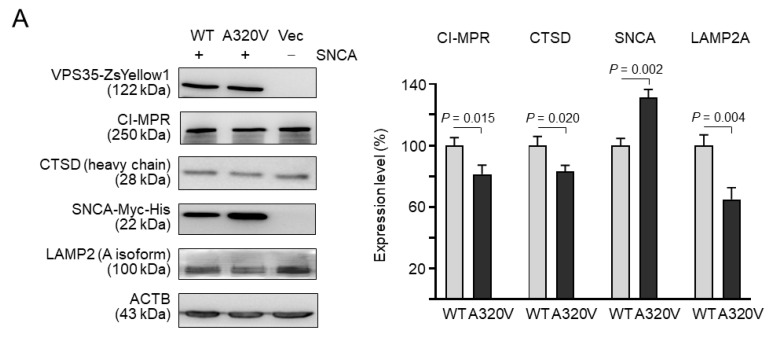
Impaired endosome-to-Golgi retrieval of cation-independent mannose-6-phosphate receptor (CI-MPR) in A320V-expressing cells. (**A**) Human embryonic kidney (HEK)-293T cells were transiently cotransfected with wild-type (WT) or mutant (A320V) *VPS35*-ZsYellow1 and α-synuclein (SNCA)-Myc-His cDNAs. The levels of exogenously expressed *VPS35*-ZsYellow1 and SNCA and endogenous CI-MPR, cathepsin D (CTSD), and lysosomal-associated membrane protein 2 (LAMP2A) were examined using Western blot analysis. Representative immunoblots from three independent experiments are shown. The densitometric quantification of CI-MPR, CTSD, SNCA, and LAMP2A versus β-actin (ACTB) is presented in the right panel. Data are expressed as the mean ± standard deviation. (**B**) Expression of *VPS35* and CI-MPR proteins was examined by confocal microscopy and colocalization analysis. Wild-type or mutant *VPS35* was labeled with green, while CI-MPR was stained with red by Cy5 fluorescence. Nuclei were counterstained with 4′-6-diamidino-2-phenylindole (DAPI; blue). The graph below represent quantification of colocalization with the means of three independent experiments.

**Figure 3 brainsci-10-00783-f003:**
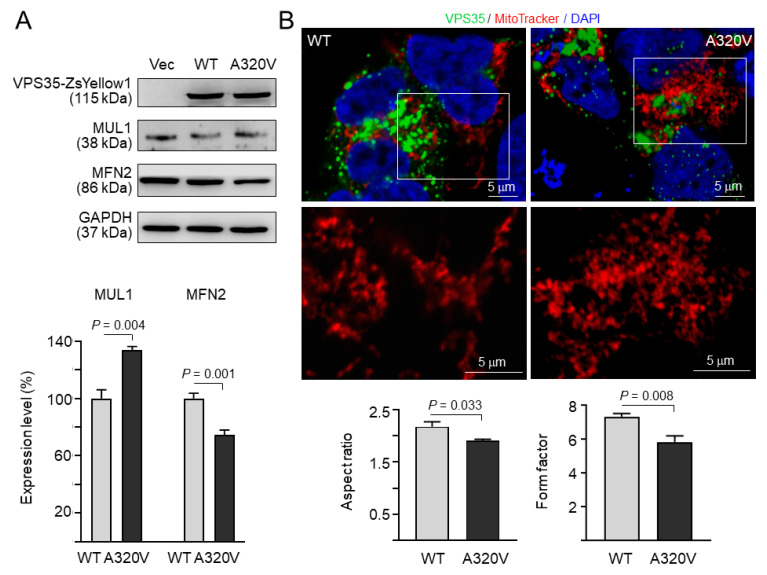
Increased MUL1 and reduced MFN2 in A320V-expressing cells. (**A**) HEK-293T cells were transiently transfected with WT or mutant (A320V) *VPS35*-ZsYellow1 cDNA. The levels of exogenously expressed *VPS35*-ZsYellow1 and endogenous MUL1 and MFN2 were examined using Western blot analysis. Representative immunoblots from three independent experiments are shown. The densitometric quantification of MUL1 and MFN2 versus glyceraldehyde-3-phosphate dehydrogenase (GAPDH) is presented in the right panel. Data are expressed as the mean ± standard deviation. (**B**) Mitochondrial morphology was examined by confocal microscopy. Wild-type or mutant *VPS35* was labeled with green, while mitochondria were stained with red by MitoTracker Deep Red FM fluorescence. Nuclei were counterstained with DAPI (blue). The graphs below represent average values of aspect ratio and form factor from three independent experiments.

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
