# Peer review of "Rare VPS35 A320V Variant in Taiwanese Parkinson’s Disease Indicates Disrupted CI-MPR Sorting and Impaired Mitochondrial Morphology"

_brainsci, 2020, doi:10.3390/brainsci10110783_

Round 1

Reviewer 1 Report

  1. Abbreviations should be explained on their first use itself in case of CI-MPR where its abbreviated form comes before the full form in the abstract.
  2. Can the authors justify selecting a mutation in the VPS35 gene that was found only in 2 patients out of a cohort of 560 PD patients for their study in terms of its significance, especially when they mention in the discussion section that the earlier D260N mutation of VPS35, although found in Taiwanese PD patients, is actually not important?
  3. Can the authors provide the raw images for the western blots of CI-MPR and CTSD since it’s difficult to relate the statistical significance depicted in the graph with the image of the blot alongside?
  4. Scale bars should be mentioned in all the confocal microscopy images.
  5. The authors are requested to provide the raw images for the blots of MFN-2 and VPS35-ZsYellow1.
  6. In the images representing mitochondrial damage via mitotracker staining the expression of VPS35 in the mutant form seems to be reduced as compared to wild type. Were the individual settings for the wild type and mutant images not kept identical here? If not, can the authors provide individual images at every chosen emission wavelength?

Author Response

  1. Abbreviations should be explained on their first use itself in case of CI-MPR where its abbreviated form comes before the full form in the abstract.

Response: We explained abbreviation CI-MPR on the first use in line 21, page 1.

  1. Can the authors justify selecting a mutation in the VPS35 gene that was found only in 2 patients out of a cohort of 560 PD patients for their study in terms of its significance, especially when they mention in the discussion section that the earlier D260N mutation of VPS35, although found in Taiwanese PD patients, is actually not important?

Response: We revised the sentence “Our results suggested that VPS35 D620N may not be an important mutation in Taiwanese PD.” In line 274, page 8: Our results suggested that VPS35 mutation may not be a common cause of Taiwanese PD.

  1. Can the authors provide the raw images for the western blots of CI-MPR and CTSD since it’s difficult to relate the statistical significance depicted in the graph with the image of the blot alongside?

Response: We provided raw images as requested.

  1. Scale bars should be mentioned in all the confocal microscopy images.

Response: We mentioned Scale bars in all the confocal microscopy images.

  1. The authors are requested to provide the raw images for the blots of MFN-2 and VPS35-ZsYellow1.

Response: We provided raw images as requested.

  1. In the images representing mitochondrial damage via mitotracker staining the expression of VPS35 in the mutant form seems to be reduced as compared to wild type. Were the individual settings for the wild type and mutant images not kept identical here? If not, can the authors provide individual images at every chosen emission wavelength?

Response: All the settings were kept identical in Fig. 3B. MitoTracker staining of mitochondria revealed small and round mitochondria, indicative of mitochondrial fragmentation, but not VPS35 expression in A320V expressing cells.

Reviewer 2 Report

The article "Rare VPS35 A320V Variant in Taiwanese Parkinson’s Disease Indicates Disrupted CI-MPR Sorting and Impaired Mitochondrial Morphology" by Wu and group explains the importance of VPS35 A320V variant that contributes to the pathophysiology of VPS35-induced PD.

The present manuscript proposes the role of vacuolar protein sorting 35 (aka VPS35) variant in Parkinson's disease. The variation in the sequence for VPS35 is not clearly known and thus the present work makes it help understand how the sequence variation is important in the pathophysiology of PD.
Although the role of VPS35 mutation in Parkinson's disease is already reported and wide research is being carried out by several groups, the present study underscores the expression of A320V mutant in the 293T cells and its further association with cation-independent mannose-6-phosphate receptor (CI-MPR), mitochondrial E3 ubiquitin-protein ligase 1 (MUL1) and alpha-synuclein. I believe the study focuses on the importance of VPS35 A320V mutation in the PD patients in the Taiwanese population which may help to develop target specific theparies.

It is a well-written manuscript, and is easy to understand the experiments conducted by the authors. Based on the results for mutation, blots, and expression, I believe the authors have discussed and concluded the argument in a decent way. The authors have proposed a question that the VPS35 A320V mutation leads to impaired mitochondrial morphology and also disrupts the CI-MPR. The question is addressed by studying the expression of A320V mutant in 293T cells, its co-localization with VPS35 and CI-MPR. Also, the altered expression LAMP2, MUL1, MFN2 and alpha-synuclein support the study to answer the raised question.

Author Response

Thank you for the positive comments.